# Age-Dependent Biomarkers for Prediction of In-Hospital Mortality in COVID-19 Patients

**DOI:** 10.3390/jcm11102682

**Published:** 2022-05-10

**Authors:** Eugene Feigin, Tal Levinson, Asaf Wasserman, Shani Shenhar-Tsarfaty, Shlomo Berliner, Tomer Ziv-Baran

**Affiliations:** 1Departments of Internal Medicine D and E, Tel Aviv Sourasky Medical Center Affiliated to the Sackler Faculty of Medicine, Tel Aviv University, Tel Aviv 6423906, Israel; tallevinson32@gmail.com (T.L.); asafw@tlvmc.gov.il (A.W.); shanis@tlvmc.gov.il (S.S.-T.); berliners@tlvmc.gov.il (S.B.); 2Infectious Diseases Unit, Tel Aviv Sourasky Medical Center Affiliated to the Sackler Faculty of Medicine, Tel Aviv University, Tel Aviv 6423906, Israel; 3Department of Epidemiology and Preventive Medicine, School of Public Health, Sackler Faculty of Medicine, Tel Aviv University, Tel Aviv 6997801, Israel; ziv.bar.cons@gmail.com

**Keywords:** COVID-19, in-hospital mortality, C-reactive protein, white blood cells, neutrophil-to-lymphocyte ratio

## Abstract

Background: Several biomarkers and models have been proposed to predict in-hospital mortality among COVID-19 patients. However, these studies have not examined the association in sub-populations. The present study aimed to identify the association between the two most common inflammatory biomarkers in the emergency department and in-hospital mortality in subgroups of patients. Methods: A historical cohort study of adult patients who were admitted to acute-care hospital between March and December 2020 and had a diagnosis of COVID-19 infection. Data on age, sex, Charlson comorbidity index, white blood cell (WBC) count, C-reactive protein (CRP), and in-hospital mortality were collected. Discrimination ability of each biomarker was observed and the CHAID method was used to identify the association in subgroups of patients. Results: Overall, 762 patients (median age 70.9 years, 59.7% males) were included in the study. Of them, 25.1% died during hospitalization. In-hospital mortality was associated with higher CRP (median 138 mg/L vs. 85 mg/L, *p* < 0.001), higher WBC count (median 8.5 vs. 6.6 K/µL, *p* < 0.001), and higher neutrophil-to-lymphocyte ratio (NLR) (median 9.2 vs. 5.4, *p* < 0.001). The area under the ROC curve was similar among all biomarkers (WBC 0.643, NLR 0.677, CRP 0.646, *p* > 0.1 for all comparisons). The CHAID method revealed that WBC count was associated with in-hospital mortality in patients aged 43.1–66.0 years (<11 K/µL: 10.1% vs. 11+ K/µL: 27.9%), NLR in patients aged 66.1–80 years (≤8: 15.7%, >8: 43.3%), and CRP in patients aged 80.1+ years (≤47 mg/L: 18.8%, 47.1–149 mg/L: 43.1%, and 149.1+: 71.7% mortality). Conclusions: WBC, NLR, and CRP present similar discrimination abilities. However, each biomarker should be considered as a predictor for in-hospital mortality in different age groups.

## 1. Introduction

The coronavirus disease 2019 (COVID-19) was identified at the end of 2019, and in March 2020 was declared a global pandemic by the World Health Organization (WHO) [1]. This pandemic has placed an unprecedented burden on healthcare systems around the world and overwhelmed many [2], putting young and experienced medical professionals alike into an unknown and challenging effort, to treat masses of patients with a new and deadly disease. Although the genetic structure of this novel virus is known [3], and data on the signs and symptoms of the disease are shared efficiently in the medical community worldwide [4], to this day we are still facing a broad community spread, many severe cases, and high morbidity and mortality. Mortality rates vary between countries [5,6,7,8], and depend on the workload in hospitals [8]. In Israel, the overall mortality rate was around 0.26% [7], and in-hospital mortality was far higher at around 22% [9].

Several studies have examined predictors of disease severity and hospitalization outcomes. These factors included demographic characteristics, background diseases, and laboratory tests. Male sex [10,11] and older age [10,11,12] were significant predictors of severe illness and mortality in various studies. Comorbidities such as chronic obstructive pulmonary disease (COPD) and diabetes mellitus were also independently associated with worse outcomes [12]. Several biomarkers were found to be associated with adverse outcomes of COVID-19 infections. These biomarkers included complete blood count parameters (lymphopenia: OR 3.33; thrombocytopenia: OR 2.36), D-dimer (OR 3.39), procalcitonin (OR 6.33), creatinine kinase (OR 2.42), aspartate transaminase (AST, OR 2.75), alanine transaminase (ALT, OR 1.71), creatinine (OR 2.84), lactate dehydrogenase (LDH, OR 5.48), and C-reactive protein (CRP, OR 4.37) [13].

A lymphocyte percentage-time model, LDH, lymphocytes, and high sensitivity CRP were found to be predictors of adverse outcomes [12].

Several studies demonstrated the usefulness of CRP and blood count as prognostic biomarkers in patients who were admitted to the hospital with COVID-19 infection [10,11,12,13,14,15,16,17,18,19,20,21,22,23,24]. However, all of these studies referred to the adult population as one group and did not examine the association between these biomarkers and adverse outcomes in sub-populations.

Therefore, the study aimed to assess whether the two most commonly used inflammatory biomarkers, taken upon admission to the emergency room, are differently associated with in-hospital mortality in subgroups of patients.

## 2. Methods

### 2.1. Study Design and Population

We examined a historical cohort study of adult patients (age ≥18 years) admitted between March and December 2020 to Tel-Aviv Sourasky Medical Center, a tertiary university-affiliated 1170-bed acute care hospital located in the center of Israel, with the diagnosis of COVID-19 infection. COVID-19 infection was diagnosed by a polymerase chain reaction (PCR) test.

### 2.2. Data Source, Measurements, and Variables

Data were obtained using MDClone (mdclone.com), a query tool that provides comprehensive patient-level data of wide-ranging variables in a defined timeframe around an index event. The MDClone system is a patient-level data extraction system designed to ease the query of large medical records [25]. Each patient admitted to the hospital upon arrival to the emergency department (ED) underwent basic inflammatory biomarkers assessment including complete blood count (CBC) and C-reactive protein.

Age, gender, comorbidities including chronic obstructive pulmonary disease (COPD), congestive heart failure (CHF), diabetes mellitus (DM), Charlson comorbidity index [26], first complete blood count (CBC), and first CRP test, were obtained. The neutrophil-to-lymphocyte ratio (NLR) was calculated. All-cause in-hospital mortality was used as the study outcome.

### 2.3. Laboratory Methods

Blood samples were collected by registered nurses or registered blood technicians using vacuum plastic tubes (spray dried K3EDTA vacuum tube and serum gel separator). Complete blood count was analyzed using DxH800 Beckman Coulter CBC analyzers (Brea, CA, USA). Wide-range CRP in human serum was analyzed by an immunoturbidimetric assay with the Siemens ADVIA 2400 chemistry system using a dedicated reagent (SIEMENS Healthcare Diagnostics Inc., Tarrytown, NY, USA) [27].

### 2.4. Statistical Analysis

Categorical variables were described using frequency and percentage. Continuous variables were evaluated for normal distribution using histograms and QQ plots and reported as median and interquartile range (IQR). A chi-square test was used to compare categorical variables between those who survived and those who died. A Mann–Whitney test was applied to compare continuous variables according to survival status. The area under the receiver operating characteristic (ROC) curve (AUC) was used to evaluate the discrimination ability of each biomarker. A Delong test was used to compare the areas under the ROC curve. The chi-square automatic interaction detection (CHAID) method was applied to identify the association between CRP, WBC, and NLR and in-hospital mortality in subgroups of patients. It classifies patients into groups using values of independent variables according to the dependent variable. This algorithm uses the well-known chi-square test to determine the significance level and to identify the independent variables with the strongest association with the studied outcome. The Bonferroni method was used to adjust the probability level. This method may merge categories of categorical variables and split continuous variables into categories. The continuous variables are binned into a predefined number of equidistant intervals. In this study, the default number was used (10 intervals). The algorithm partitions the patients into two or more child nodes and continues with repeated partitions of each subset of patients until a stopping criterion is satisfied [28]. All statistical tests were two-tailed and a *p*-value lower than 0.05 was considered statistically significant. All statistical analyses were performed using SPSS (IBM SPSS Statistics for Windows, version 27, IBM Corp., Armonk, NY, USA, 2020).

## 3. Results

Seven hundred and sixty-two patients were included in the study. Of them, 60% were male and the median age was 70.9 years. Patients’ characteristics are presented in Table 1.

Twenty-five percent of the patients died during hospitalization. A comparison between those who died and those who survived is presented in Table 2. Older age (median 79.8 vs. 67.6 years, *p* < 0.001), male gender (27.1% vs. 21.2%, *p* = 0.042), higher Charlson score (median 5 vs. 3, *p* < 0.001), higher CRP (median 138 mg/L vs. 85 mg/L, *p* < 0.001), higher WBC count (median 8.5 vs. 6.6 K/µL, *p* < 0.001), and higher NLR (median 9.2 vs. 5.4, *p* < 0.001) were associated with in-hospital mortality.

The discrimination ability of wrCRP, WBC count, and NLR was similar (AUC: 0.646, 95% CI 0.601–0.690; 0.643, 95% CI 0.593–0.690; 0.677, and 95% CI 0.631–0.723, respectively) with no significant difference between them (*p* > 0.05 for all comparisons, Figure 1).

CHAID analysis was applied to study the association between specific inflammatory biomarkers and in-hospital mortality in sub-groups of patients (Figure 2). The analysis demonstrated that WBC count was associated with in-hospital mortality in patients aged 43.1–66.0 years (<11 K/µL: 10.1%, ≥11 K/µL: 27.9%, *p* = 0.002), NLR in patients aged 66.1–80.0 years (≤8 K/µL: 15.7%, >8 K/µL: 43.3%, *p* < 0.001), and CRP in older patients (≤47.0 mg/L: 18.8%, 47.1–149 mg/L: 43.1%, >149 mg/L: 71.7%, *p* < 0.001). A very low mortality rate (1.4%) was observed in younger patients (≤43) and none of the biomarkers were associated with increased mortality in this age group.

## 4. Discussion

The current study has demonstrated the effectiveness of CRP, NLR, and WBC as predictors for in-hospital mortality among COVID-19 patients.

The study has shown that while the ability of these biomarkers to predict in-hospital mortality in all adult patients as one group is similar, it may be preferable to use each biomarker in different age groups. Hence, younger patients (43–66 years) will benefit from WBC as the prognostic marker of choice while the older patients (66–80 years) will benefit from NLR, and octogenarians and older (>80 years) will benefit from CRP. Previous studies showed that patients with comorbidities were at increased risk for in-hospital mortality [10,11,12]. Therefore, the current study assumed that a higher Charlson comorbidity index is associated with increased risk for in-hospital mortality, which was demonstrated in the univariate analysis. However, in the CHAID analysis, the Charlson comorbidity index was not included as a discriminating factor of the probability of in-hospital mortality. This finding can be explained by the fact that older age was associated with increased in-hospital mortality as well as with a higher index. Since one point is given for every decade from the age of 50 and over (maximum of four points), it can partially explain the absence of the Charlson comorbidity index in the CHAID analysis. Other possible explanations include that patients delayed access to treatment and thus worsened their condition, resulting in a higher probability of death [29].

It is a well-established fact that the immune system acts and reacts differently as people age. Thus, it is not surprising that different biomarkers predict mortality better in different age groups. In older patients, there is an increase in interleukin 6 (IL-6) and a decrease in neutrophil survival in response to stimuli [30]. This further supports the claim that CRP is the superior predictor in the elderly being downstream from IL-6 [31]. Indeed, this specific biomarker is a powerful prognostic predictor in patients with COVID-19 infection that highly correlates with other relevant inflammatory biomarkers [11,12,18,23,24], suggesting a relatively effective and low-cost tool that is available in real-time to the medical team.

The concept of the prognostic utility of CRP measurements in respiratory viral infections is not new. Zimmerman et al. described the prognostic value of the first CRP measurement of patients hospitalized due to H1N1 influenza [32]. Inflammatory biomarkers in general, and specifically CRP, are known to have a prognostic value in COVID-19 patients [10,11,12,13,14,15,16,17,18,19,20,21,22,23,24]. The availability of the CRP and CBC, in comparison to other biomarkers, is even more pronounced when considering that both are available as a point-of-care test that can be completed at the clinic or at the nursing home where the patients reside, at their bedside [33,34,35]. Due to their availability at relatively low costs, they could be used in retirement homes, community clinics, and home visits by general practitioners and nurses for risk stratification and deciding whether to refer patients to further treatment in a hospital.

### Limitations

The current study had several limitations.

First, this is a retrospective, single-center medical records-based study. Second, a small number of laboratory tests were not available in the information system, probably due to technical issues. Third, the study included only hospitalized patients and therefore represents the most severe or difficult-to-treat COVID-19 patients. The mortality rates observed in this cohort are similar to those reported in other studies performed on hospitalized patients in Israel. Fourth, we focused on CBC and wrCRP as our inflammatory biomarkers of choice since they are routinely taken and relatively available both in hospitals and in community practices. Other biomarkers are also available and may be used to classify patients better. Fifth, corticosteroids are used for treatment of severe COVID-19 [36] and may increase the absolute neutrophil count and WBC count. However, this treatment is given regularly only to hospitalized COVID-19 patients and since all blood tests were taken upon admission to the ER it does not influence the study results. Sixth, the study included only patients who were hospitalized before a vaccine was available. The vaccination program is not the same for all ages and the inclusion of patients that were hospitalized before a vaccine was proposed allowed us to examine the association between the biomarkers and in-hospital mortality without the confounding effect of the vaccine.

## 5. Conclusions

WBC, NLR, and CRP present a similar ability to discriminate between patients who die in the hospital and those who do not. However, each biomarker should be considered as a predictor for in-hospital mortality in different age groups. Thus, age and specific laboratory data upon arrival to the ER or to the community practice should be considered for referral of a new COVID-19 patient to the internal medicine department, ICU, or ambulatory care.

## Figures and Tables

**Figure 1 jcm-11-02682-f001:**
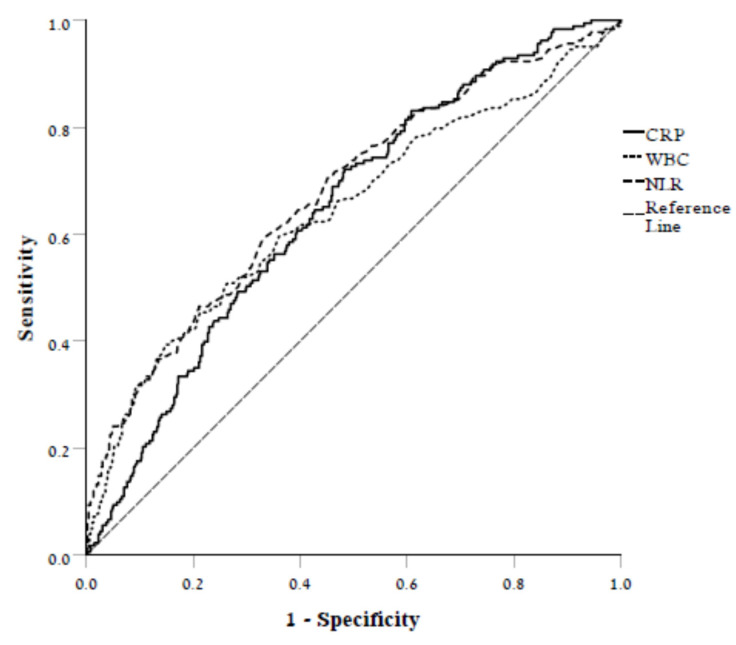
Receiver operating characteristic curve illustrating the ability of CRP, WBC, and NLR to discriminate between those who died and those who survived.

**Figure 2 jcm-11-02682-f002:**
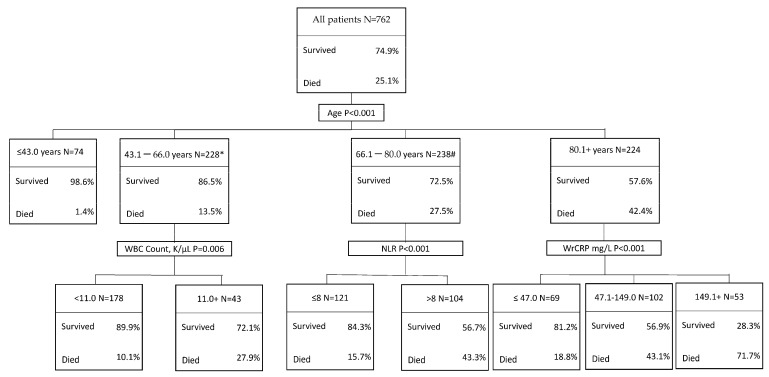
Classification tree using the CHAID methodology showing the association between in-hospital mortality and WBC, NLR, and CRP in subgroups of patients. * 7 patients had missing CBC count, # 11 patients had missing CBC count.

**Table 1 jcm-11-02682-t001:** Patient’s characteristics.

Parameter	All (*N* = 762)
Age (years), median (IQR)	70.9 (57.8–82.0)
Male, n (%)	455 (59.7%)
COPD, n (%)	76 (12.4%)
CHF, n (%)	52 (8.5%)
DM, n (%)	203 (33.1%)
Charlson comorbidity index, median (IQR)	4 (2–5)
CRP (mg/L), median (IQR)	95.4 (36.9–155.6)
WBC count (K/µL), median (IQR)	6.9 (5.1–9.8)
NLR, median (IQR)	6.35 (3.40–11.30)

IQR—interquartile range; COPD—chronic obstructive pulmonary disease; CHF—congestive heart failure; DM—diabetes mellitus; CRP—C-reactive protein; NLR—neutrophil-to-lymphocyte ratio; WBC—white blood cell.

**Table 2 jcm-11-02682-t002:** Univariate analysis of the predictors for in-hospital mortality.

Parameter	In-Hospital Mortality		*p*
	Survived (*N* = 571)	Died (*N* = 191)	
Age (years), median (IQR)	67.6 (53.7–78.9)	79.8 (70.5–87.2)	<0.001
Male, n (%)	329 (57.6%)	126 (66.0%)	0.042
COPD, n (%)	59 (12.7%)	17 (11.4%)	0.680
CHF, n (%)	35 (7.5%)	17 (11.4%)	0.139
DM, n (%)	147 (31.6%)	56 (37.6%)	0.178
Charlson comorbidity index, median (IQR)	3 (2–5)	5 (4–7)	<0.001
CRP (mg/L), median (IQR)	84.9 (28.9–149.5)	138.2 (69.0–187.6)	<0.001
WBC count (K/µL), median (IQR)	6.6 (5.0–8.8)	8.5 (5.9–12.8)	<0.001
NLR, median (IQR)	5.4 (3.0–9.8)	9.2 (5.2–19.3)	<0.001

CHF—congestive heart failure; COPD—chronic obstructive pulmonary disease; DM—diabetes mellitus; IQR—interquartile range; NLR—neutrophil-to-lymphocyte ratio; WBC—white blood cell; CRP—C-reactive protein.

## Data Availability

The data is kept in the MDClone server in Tel Aviv Sourasky Medical Center and is not available for distribution due to patient confidentiality regulations within the organization.

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
