# Peer review of "Age-Dependent Biomarkers for Prediction of In-Hospital Mortality in COVID-19 Patients"

_jcm, 2022, doi:10.3390/jcm11102682_

Round 1

Reviewer 1 Report

The authors report an extensive evaluation of common and readily available inflammatory markers in single-centre sample of COVID-19 patients in the beginning of the epidemic (Mar-Dec 2020). They found differences in the predictivity of each biomarkers in different age groups, an interesing issue not widely explored even after two years of pandemic. Nonetheless, a revisions is advisable in different parts of the manuscript

  • spelling mistakes need to be corrected (see Introduction line 4 page 1)
  • language should be revised to be more fluent and direct
  • among limitations it should be noted that data have collected in unvaccinated population and before any treatment (both antiviral and immunomodulating) become available. Regarding the latter issue, considering the period in which patients have been analyzed it is expected that for at least a half of the observation time-frame steroid was administered. Moreover, no data is shown regarding the immune status of the subjects, which normally strongly influences disease evolution.

In conclusion, the manuscript has some novelty and the study tried to find  easy-to-use biomarkers to be suitable for different ages, but data presentation and patients characterization should be implemented.

Author Response

We thank you for considering our manuscript to be published in the Journal of Clinical Medicine.

We have carefully read the reviewers’ recommendations for the improvement of the paper and have changed the manuscript accordingly.

All comments were addressed.

Thank you

We thank the reviewer for his constructive comments.

  1. R: Spelling mistakes need to be corrected (see Introduction line 4 page 1). Language should be revised to be more fluent and direct.

A: Thanks for bringing up the subject. The manuscript has been linguistically edited.

  1. R: Among limitations it should be noted that data have collected in unvaccinated population and before any treatment (both antiviral and immunomodulating) become available. Regarding the latter issue, considering the period in which patients have been analyzed it is expected that for at least a half of the observation time-frame steroid was administered. Moreover, no data is shown regarding the immune status of the subjects, which normally strongly influences disease evolution.

A: We agree with this comment. In this study, all blood tests were taken upon emergency department admission. In Israel, treatment with steroid was only available hospitalized patients suffering severe COVID-19. The limitation paragraph has been expanded to include the issues raised.

Reviewer 2 Report

The authors aim to highlight the interdependent role of biomarkers and age in in-hospital mortality from COVID-19. The document could be interesting but both the objectives and the results should be discussed taking more fully into consideration the literature that has now been produced on these topics.

Major revisions

- The introduction is inadequate. Two years after the start of the pandemic, there have been many works that have examined the role of age in Covid mortality, also in relation to biomarkers. A broader and more detailed exposition of the existing literature is therefore necessary to justify the objectives of this further paper.

- Consequently, the discussion should also be considerably broadened

- In CHAID analysis, nominal, ordinal, and continuous data can be used, where continuous predictors are split into categories with approximately equal number of observations. Explain if the continuous variables have been categorized. If so, indicate which cut-offs were used; if not, explain why and if the generated groups are numerically homogeneous

- In the chad method, a correct Bonferroni p value must be calculated due to the steps that create the crosstabs. Instead, in the methods the authors did not take into account this important correction which could modify the reported results in terms of significance.

Minor revisions

- Add the numerosity (n = ..) in the column headings of tables 1 and 2 and in the nodes of the Classification tree

Author Response

We thank you for considering our manuscript to be published in the Journal of Clinical Medicine.

We have carefully read the reviewers’ recommendations for the improvement of the paper and have changed the manuscript accordingly.

All comments were addressed.

Thank you

We thank the reviewer for his constructive comments.

  1. R: The introduction is inadequate. Two years after the start of the pandemic, there have been many works that have examined the role of age in Covid mortality, also in relation to biomarkers. A broader and more detailed exposition of the existing literature is therefore necessary to justify the objectives of this further paper. Consequently, the discussion should also be considerably broadened.

A: The introduction has been updated and further references were added.

  1. R: In CHAID analysis, nominal, ordinal, and continuous data can be used, where continuous predictors are split into categories with approximately equal number of observations. Explain if the continuous variables have been categorized. If so, indicate which cut-offs were used; if not, explain why and if the generated groups are numerically homogeneous. In the CHAID method, a correct Bonferroni p value must be calculated due to the steps that create the crosstabs. Instead, in the methods the authors did not take into account this important correction which could modify the reported results in terms of significance.

A: The statistical method paragraph has been expanded to include a detailed explanation and p-values were updated according to Bonferroni method.

  1. R: Add the numerosity (n =..) in the column headings of tables 1 and 2 and in the nodes of the Classification tree.

A: The numbers of patients were added in table 1, table 2 and in figure 2.

Round 2

Reviewer 1 Report

The manuscript has been reviewed addressing the major issues. Even though the study has several limitations, they have been discussed and the manuscript bring some novelty, especially in trying to approaching patients with COVID-19 considering their age

Author Response

We thank the reviewer for the constructive comments.

Reviewer 2 Report

The authors have accepted the suggestions by improving their manuscript.

Minor revision

As the authors verified, in the univariate analysis the high values ​​of the Charlson comorbidity index were strongly associated with higher in-hospital mortality. This was quite predictable. However, after the CHAID analysis, the CCI is no longer a discriminating factor of the probability of dying in hospital; this apparently seems inexplicable given that even the literature highlights the role of co-morbidities on events. The authors should discuss and justify this finding by taking for example a cue from the recent work by Bartolomeo et al. [Bartolomeo, N.; Giotta, M.; Trerotoli, P. In-Hospital Mortality in Non-COVID-19-Related Diseases before and during the Pandemic: A Regional Retrospective Study. Int. J. Environ. Res. Public Health 2021, 18, 10886. https://doi.org/10.3390/ijerph182010886], who demonstrated a significant increase during the pandemic in in-hospital mortality in NO-Covid patients with higher CCI. Therefore, probably during the pandemic, the most fragile subjects (with higher CCI) protected themselves from the infection of the SARS-CoV-2 virus but delayed access to treatment worsened their condition resulting in a higher probability of death in hospital due to causes other than COVID19.

Author Response

We thank the reviewer for his comment and have added an explanation in the discussion section
